# Delta Opioid Peptide Targets Brain Microvascular Endothelial Cells Reducing Apoptosis to Relieve Hypoxia-Ischemic/Reperfusion Injury

**DOI:** 10.3390/pharmaceutics15010046

**Published:** 2022-12-23

**Authors:** Ran Zhang, Meixuan Chen, Zhongfang Deng, Lingchao Kong, Bing Shen, Lesha Zhang

**Affiliations:** 1Department of Physiology, School of Basic Medical Sciences, Anhui Medical University, Hefei 230032, China; 2Department of Orthopedics, The First Affiliated Hospital of Anhui Medical University, Hefei 230032, China

**Keywords:** δOR, BMECs, ischemia, OGD/R, apoptosis

## Abstract

Stroke is one of the leading causes of death. (D-ala2, D-leu5) enkephalin (DADLE) is a synthetic peptide and highly selective delta opioid receptor (δOR) agonist that has exhibited protective properties in ischemia. However, the specific target and mechanism are still unclear. The present study explores the expression of δOR on brain microvascular endothelial cells (BMECs) and whether DADLE could relieve I/R-induced injury by reducing apoptosis. A lateral ventricular injection of DADLE for pretreatment, the neurofunctional behavior score, and TTC staining, were used to evaluate the protective effect of DADLE. Immunofluorescence technology was used to label different types of cells with apoptosis-positive signals to test co-localization status. Primary cultured BMECs were separated and treated with DADLE, accompanied by OGD/R. The CCK-8 test was conducted to evaluate cell viability and TdT-mediated dUTP Nick-end Labelling (TUNEL) staining to test apoptosis levels. The levels of apoptosis-related proteins were analyzed by Western blotting. The co-localization results showed that BMECs, but not astrocytes, microglia, or neurons, presented mostly TUNEL-positive signals, especially in the Dentate gyrus (DG) area of the hippocampus. Either activation of δORs on rats’ brains or primary BMECs mainly reduce cellular apoptosis and relieve the injury. Interference with the expression δOR could block this effect. DADLE also significantly increased levels of Bcl-2 and reduced levels of Bax. δOR’s expressions can be detected on the BMECs, but not on the HEK293 cells, by Western blotting and IFC. Therefore, DADLE exerts a cytoprotective effect, primarily under hypoxia-ischemic injury/reperfusion conditions, by targeting BMECs to inhibit apoptosis.

## 1. Introduction

Strokes severely threaten public health and remain one of the top ten lethal diseases worldwide [1]. In total, 87 percent of strokes happen to result from thrombus-induced local cerebral ischemia [2]. Patients who suffer from an ischemic stroke often receive thrombolytic therapy in clinics [3], which potentially leads to reperfusion injuries, which induces cellular apoptosis and irreversible neuronal death [4,5]; this causes damage to sensory, motor, and even memory [6] functions. If the tissue-protective agent could be applied before or during the procedure of thrombolytic therapy, the ischemia/reperfusion (I/R) injury might be relieved, so it is critical to discover tissue-protective drugs. 

One of the G-protein coupled receptors family, the delta opioid receptor (δOR), mainly takes part in the regulation of analgesia, emotion, and learning [7]. Although many researchers in the last decade have reported that several artificially synthetic and specific agonists of δOR possess a tissue-protective property against ischemia [8,9,10], the mechanism and direct target are still not fully known. Among these, the peptide [D-ala2, D-leu5) enkephalin (DADLE) has been shown to act upon primary cultured neurons [11,12], or cell lines, derived from the central neural system (CNS), and reduce neural apoptosis in vitro [13]. Other evidence indicates that DADLE can induce autophagy and inhibit apoptosis in astrocytes [12]. However, whether DADLE acts directly upon neurons has not been confirmed in vivo. It has been reported that δOR is involved in the regulation of vascular function [14]. For example, delta agonists decrease the rate of endothelin release from cultured porcine aortic endothelial cells [15]. Additionally, δORs have been demonstrated to have a functional presence in vascular smooth muscle [16]. The δOR agonist [D-Pen2, D-Pen5], enkephalin (DPDPE), has been shown to have no effect on resting tension; however, it enhances the contractions induced by noradrenaline on rat aortic vascular smooth muscle [17]. There is no evidence supporting the expression of δOR on the cerebral vascular endothelial cells. However, in this study we first found that when the classical model Middle Cerebral Artery Occlusion and Reperfusion (MCAO/R) was applied to mimic the I/R pathophysiological process, most of the cellular apoptosis in the hippocampus, especially the Dentate gyrus (DG) area, appeared in the cerebral vascular endothelial cells, but not the neuron or glia. Based on the above, we supposed whether delta opioid peptide may target brain vascular endothelial cells, reducing apoptosis to relieve hypoxia-ischemic/reperfusion injury. As relieving vascular endothelial cells’ injury suffering from I/R is beneficial for maintaining the blood supply and integrity of the blood–brain barrier [18,19], this study may be helpful in exploring the mechanisms of tissue-protective agents against ischemic stroke. 

## 2. Materials and Methods

### 2.1. Animal Treatment

Sprague Dawley male rats aged 7–8 weeks and weighing 250–280 g, purchased from the animal center of Anhui Medical University (Hefei, Anhui, China), were housed in 485 mm × 350 mm × 200 mm cages (2–3 rats per cage) and were maintained on a 12 h light/dark cycle at room temperature (RT, ~25 °C), ad libitum. All studies involving animals complied with the ARRIVE guidelines [20]. In addition, all experiments were approved by the Administration Office of Laboratory Animals of Anhui Medical University and permitted by the Animal Ethics Committee of Anhui Medical University (License No. LLSC20190533). All the rats in each group for each trial in the experiments were randomly selected.

### 2.2. Intracerebroventricular Administration

Rats were anaesthetized with Zoletil—tiletamine/zolazepam (50 mg/kg, Zoletil®, Virbac, France; Zoletil powder diluted with 5 mL sterile water for injection; 50 mg/mL tiletamine/50 mg/mL zolazepam) by intraperitoneal injection [21,22]. The animals were then fixed on a stereotaxic apparatus (RWD Life Science, Shenzhen, China). To avoid the covering of the skulls on the right side, thus disturbing the cerebral blood flow (CBF) monitoring, guide cannulas were implanted on the unilateral left side of the rat brain (AP: −0.8 mm; ML: +1.5 mm; and DV: −2.0 mm). Each cannula had a stainless-steel blocker to keep them patent, and 80,000 IU penicillin was subcutaneously injected to prevent infection. Rats were single and separately housed in 370 mm × 270 mm × 170 mm cages for recovery for seven days. Artificial cerebrospinal fluid (ACSF), or the δOR agonist DADLE, (E7131, Sigma-Aldrich,St. Louis, MO, USA) dissolved in ACSF at a concentration of 5 nmol/10 μL, were intracerebroventricularly administrated to rats through a 31-gauge injection cannula (1.8 mm under the tip of the guide cannula) over 5 min at a rate of 2 μL/min and with an additional 1 min for drug diffusion. The antagonist NTI (2.5 nmol/5 μL) was injected 30 min before the injection of DADLE in the same condition. During the microinjection, the rats were awake and gently restrained by hand. 

### 2.3. MCAO/R Model

Cerebral ischemic stroke and reperfusion injury were induced by establishing MCAO/R model [23]. Fifteen minutes after receiving DADLE or ACSF, the rats were anaesthetized, and their cerebral hemisphere was monitored by laser speckle imaging (PeriCamPSI, PERIMED, Järfälla, Sweden). A monofilament purchased from RWD Life Science (Shenzhen, China) was inserted 18 to 20 mm into the internal carotid to occlude the origin of the MCA. The time interval between drug administration and blood flow of the artery being blocked was averagely 45 min, as the previous studies described [24]. After 1 h, the monofilament was removed for reperfusion. Before recovering, the rats were placed on a heating plate. The rats in the sham group underwent the same surgical operations except for the monofilament insertion. Statistically, the total mortality was 21% [23,25]. 

### 2.4. Evaluation of Neurologic Deficit Scores

According to the Garcia criteria [26], the sensory and motor neurologic deficits induced by ischemia/reperfusion injury were evaluated 72 h after MCAO/R model establishment. This test included six items (maximal score of each item = 3, maximal sum of scores = 18): spontaneous activity (0–3 points), limb symmetry (0–3 points), forepaw outstretching (0–3 points), climbing wire cage (1–3 points), body proprioception (1–3 points), and response to vibrissae touch (1–3 points). The degree of neurologic deficit was inversely proportional to the summation of all six individual test scores. Before assessment, the examiners were unaware of which group the rats belonged to.

### 2.5. Infarct Volume Assessment

The cerebral ischemic infarct volume was observed using 2, 3, 5-triphenyltetrazolium chloride (TTC) staining. After the neurologic evaluation, the rats were anaesthetized with an overdose anesthetic (Zoletil, 120 mg/kg, i.p.) and sacrificed. The rats’ brains were rapidly removed and frozen at −20 °C for 15 min. Except for the olfactory bulb, cerebellum, and low brain stem, the brain was sliced into 2-mm-thick slices, from rostral to caudal, by using brain matrices made of stainless steel (RWD Life Science, Shenzhen, China). The coronal sections were stained with 2% TTC (T8877, Sigma-Aldrich, USA) diluted with PBS at 37 °C for 15 min per side, avoiding light. Photographic images were taken to measure the stained (red) and unstained (white) areas and calculate the percentage of the infarct volume of the unstained area by using Image-Pro Plus 6.0 image software (Media Cybernetics, Rockville, MD, USA). 

### 2.6. Cell Cultures

Primary brain microvascular endothelial cells (BMECs) were separated according to a previous method [27]. Specifically, a total of six Sprague Dawley rats (aged 2–3 weeks, weighing 10–20 g, sex in half) were anesthetized by a gas mixture containing 2% isoflurane. Then, the rats were humanely killed through the inhalation of excess carbon dioxide and cleansed with 75% alcohol. Under aseptic conditions, their cerebral hemispheres were then removed and immersed in sterile phosphate-buffered saline (PBS) solution. After removing the pia mater, surface blood vessels, and the cerebral medulla, the remaining tissue was cut into pieces and homogenized into suspensions. The components that could pass a 175-μm mesh screen but be intercepted by a 75-μm mesh screen were harvested. After centrifugation at 1000 rpm for 5 min at 4 °C, the resulting precipitate was digested with 0.2% type II collagenase (Sigma-Aldrich, St. Louis, MO, USA) in PBS at 37 °C for 30 min and then subsequently centrifuged. The obtained precipitate was resuspended and cultured in Primary Endothelial Cell Complete Medium (Pricells, Hubei, China) containing 10% fetal bovine serum, 1% endothelial cell growth supplement, and 1% penicillin/streptomycin solution. HEK293 cells were purchased from Procell Life Science & Technology Corporation (Hubei, China), and cultured in MEM containing 10% FBS and 1% penicillin/streptomycin solution. The cells were incubated at 37 °C in 5% CO_2_.

### 2.7. TdT-Mediated dUTP Nick-End Labeling (TUNEL) Assay

The rats were deeply anaesthetized and perfused with saline, followed by 4% paraformaldehyde at 72 h after MCAO/R. Their brains were then removed and kept in a 30% sucrose solution to dehydrate until they sunk after five days. Then, 20-micron-thick brain slices were obtained by cryostat section (CM3050S, Leica, Germany). The primary cultured cells were seeded on coverslips for 24 h and fixed with 4% paraformaldehyde for 20 min at R.T. A TUNEL Bright Green apoptosis detection kit (A112, Vazyme Biotech, Nanjing, China) was applied. The nuclei were stained with a 4′-6-diamidino-2-phenylindole (DAPI) staining solution (Beyotime Biotechnology, Shanghai, China). Finally, the fluorescence signals were visualized with an Axio Observer 3 microscope (Carl Zeiss AG, Oberkochen, Germany), and Image-Pro Plus 6.0 was used to perform statistics.

### 2.8. Oxygen Glucose Deprivation/Reperfusion (OGD/R) and Drug Treatment

Cells were seeded in 12-well plates overnight; then, the medium was replaced with an equilibrated salt solution (EBSS) (Procell, Hubei, China) without glucose. Subsequently, the plates were placed in a hypoxic incubator (H35 HPEA, Don Whitley Scientific, Yorkshire, UK) filled with a gas mixture containing 0.1% oxygen, 5% carbon dioxide, and 94.9% N_2_ for 12 h at 37 °C. After OGD, the cells were replaced with fresh medium and incubated in an incubator at 37 °C/5% CO_2_ for reperfusion for 6 h. During the OGD/R procedure, different groups of cells were continuously exposed to solvent (vehicle group), or 5 nM DADLE, or 10 nM DADLE, or 10 nM DADLE + 1 μM NTI (Naltrindole, 9000705, Cayman Chemical, Ann Arbor, MI, USA).

### 2.9. Cell Viability Assay

Cell viability was determined by using an Enhanced Cell Counting Kit-8 (Beyotime, Shanghai, China). Cells were seeded into 96-well plates for 24 h. A CCK-8 reagent was added to each well and incubated for another 2 h. Finally, absorbance at 450 nm was recorded using a microplate reader.

### 2.10. SiRNA Transfection

The RNA oligonucleotides were synthesized by Sangon Biotech (Shanghai, China). The sequences of siRNA targeting δOR were referred to in a previous report [28] as follows: 

Sense: 5’-GGCUGUGCUCUCCAUUGACUU-3’;

Antisense 5’-GUCAAUGGAGAGCAGCCUU-3’.

The scrambled sequences were designed as a mismatch control:

Sense: 5’-GGCGUGUCUCUCUCUCGACUU-3’

Antisense: 5’-GUCGUAAGAGACACGCCUU-3’.

PolyFast Transfection Reagent (HY-K1014, MCE, NJ, USA), mixed with siRNA, was applied on the cells seeded in 12-well plates for 48 h, followed by specific experimental assays.

### 2.11. Immunofluorescence

Brain slices or the cells seeded on the coverslips were fixed with 4% PFA first, and then washed three times in PBS plus 0.3% Triton X-100 with gentle agitation, and blocked in 10% bovine serum albumin (BSA) solution in TBS for 2 h at RT. Then, the primary antibody was incubated at 4 °C overnight. Specifically, for tissue assay, the antibody of the von Willebrand Factor (vWF) (sc-365712, mouse, 1:100, Santa, Dallas, TX, USA), Iba-1 (ab153696, rabbit, 1:100, Abcam, Cambridge, UK), GFAP (ab7260, rabbit, 1; 100, Abcam, UK), or NeuN (MAB377, mouse, 1:100, EMD millipore, MA, USA) were used. For δOR identification, BMECs and HEK293 cells were incubated at 4 °C with the vWF antibody (mouse, 1:100, Santa Cruz, Dallas, TX, USA) and anti-δOR (ab176324, rabbit, 1:1000, Abcam, UK) for 12 h. A fluorescent tagged secondary (Dylight 649: goat anti-rabbit IgG 1:100 or Alexa Fluor® 555 donkey anti-Mouse IgG 1:100) antibody was incubated for 2 h at R.T. Fluorescence images were taken using the above-mentioned microscope, and Image-Pro Plus 6.0 was used to perform statistics. For the confocal laser microscopy, cells were seeded onto poly-D-lysine-coated coverslips placed in a 24-well plate. Except the fluorescent tagged secondary Dylight 649: goat anti-rabbit IgG was changed to Alexa Fluor 488-conjugated AffiniPure Donkey Anti-Rabbit IgG (1:100), the other antibodies were used in the same condition. Fluorescence was observed using a Laser confocal microscope (Leica AG, Weztlar, Germany). 

### 2.12. Western Blotting

Equal amounts of protein were electrophoresed on 12% SDS-PAGE gels and transferred to PVDF membranes. After blocking with a 5% non-fat milk dilution in PBST for 1 h at RT, the membranes were incubated with primary antibody dilutions overnight at 4 °C. The primary antibodies anti-Bax (50599-2-lg, rabbit, 1:1000, proteintech, Hubei, China), anti-Bcl-2 (26593-1-AP, rabbit, 1:1000, proteintech, Hubei, China), Anti-Caspase-9 (10380-1-AP, rabbit, 1:1000, proteintech, Hubei, China), and Anti-δOR (ab176324, rabbit, 1:1000, Abcam, UK) were used. After being rinsed with TBST solution three times, HRP-conjugated goat anti-rabbit IgG (1:5000) was incubated for 2 h at RT. Then, an ECL detective reagent (GE Health, Little Chalfont, UK) was applied. The densitometry of immunoblots was quantified using a chemiluminescence gel imaging analysis system (P&Q, Shanghai, China).

### 2.13. Statistical Analysis

Data were presented as mean ± S.E.M. and analyzed using GraphPad Prism 8.0.2.263 (San Diego, CA, USA). The statistical significance was determined by using Student’s unpaired *t*-test when only two groups were compared. When three or more groups were compared, statistical significance was determined by one-way ANOVA followed by Tukey’s Multiple Comparison Test. A two-sided value of *p* < 0.05 was considered statistically significant. A two-way ANOVA analysis was used among the grouped data.

## 3. Results

### 3.1. The Administration of Delta Opioid Peptide DADLE Reduces MCAO/R-Induced Brain Infarct Volume and Improves Neurologic Function

The MCAO/R model is a classical model that could mimic ischemic stroke and reperfusion injury in rats. Several brain areas, including the striatum, dorsal hippocampus, and sensory and motor cortex, are affected by occlusion of the cerebral middle artery [29]. Using laser speckle imaging, the infarct area, and the blood flow of the focused right side of rats’ brains could be monitored by comparing them before and after surgery (Figure 1A). At 72 h post-surgery, the sensory and motor neurologic deficits of six rats were evaluated by Garcia scores, which showed that MCAO/R-caused decreased spontaneous activity, dysfunction in climbing, outstretching, and sensation to touch. However, the pre-administering of DADLE (5 nmol/10 μL, i.c.v.) could obviously improve these neurologic deficits (Figure 1B). TTC staining also displayed that the I/R-induced infarct cerebral volume could be significantly shrunk by DADLE pre-treatment (Figure 1C). In addition, both the DADLE-induced reduction of brain infarct volume and improvement of neurologic function could be significantly inhibited by the antagonist of δOR which is NTI (Figure 1B,C). These results confirmed the neuroprotective property of DADLE against brain I/R injury.

### 3.2. MCAO/R-Induced Apoptosis Mostly Appears in Cerebral Microvascular Endothelial Cells but Not in Neurons and Glia

Although DADLE could promote the survival of neural cells in vitro [11], whether it could directly target neurons in vivo still lacks evidence, and the specific molecular mechanism is not fully known. To determine the cell type that is mainly affected by DADLE, we visualized the co-localized situation of apoptosis objects via TUNEL staining and the immunofluorescence of different cellular markers, including the marker of vascular endothelial cells, vWF, the marker of neuron, NeuN, the marker of microglia, Iba-1, and the marker of astroglia, GFAP. Representatively, we show the pictures captured from the DG area of the hippocampus in Figure 2. Among them, TUNEL-positive objects distributed in strips and streaks that merge with vWF signals, which aroused our interest. This indicates that the majority of apoptosis-occurred cells may be cerebral microvascular endothelial cells, 72 h after MCAO/R (Figure 2A). In contrast, the co-localized signals of TUNEL-positive objects with either neurons, microglia, or astroglia are much lower (Figure 2B–D). Similar phenomena emerge in the CA1 and CA2 areas of the hippocampus, as well as in the cortex, as shown in Appendix A–S3. The statistical results of the percentages of the respective cellular-type-specific apoptotic portion within the above areas are displayed in Figure 2F. Meanwhile, the injection of DADLE could reduce the apoptotic level of cerebral microvascular endothelial cells after the MCAO/R, as the TUNEL-positive objects were significantly decreased in the DADLE-applied group when compared with the ACSF-treated group (Figure 2E). The two-way ANOVA results showed that the drug factor accounted for approximately 50.55% of the total variance (F(3, 16) = 19.63, *p**** = 0.0004), showing the significance. Furthermore, our results indicated that DADLE could weaken the ischemia/reperfusion triggered neuroinflammation-related activation of microglia and astroglia [30,31], based on the fluorescence intensity of the positive Iba-1- and GFAP-signals being decreased by DADLE-administration (Appendix A). The two-way ANOVA results showed that the drug factor accounted for approximately 44.64% of the total variance (F(3, 16) = 15.84, *p*** = 0.0011) in the change of mean IF intensity of GFAP; this accounted for approximately 50.89% of the total variance (F(3, 16) = 19.01, *p**** = 0.0005) in the change of mean IF intensity of Iba-1, and both of the above showing the significance. From the above results, we speculate that MCAO/R mainly induces the apoptosis of cerebral microvascular endothelial cells at 72 h post-ischemia, and that DADLE specifically relieves it.

### 3.3. Primarily Cultured Brain Microvascular Endothelial Cells (BMECs) Express Endogenous Delta Opioid Receptors

As a member of GPCRs, delta opioid receptors (δORs) express widely from the peripheral to the central neural system. Ten years ago, the presence of functional δOR was found in vascular smooth muscles [16]. Subsequently, δORs’ expression was detected in the retina by immunohistochemistry [32] or in the thoracic aorta by western blotting [33]. However, the notion that δORs express on other vascular-derived cells has barely been reported. Now that we know that δOR agonist specifically reduces the MCAO/R-caused apoptosis of cerebral microvascular endothelial cells in vivo, it is reasonable to speculate that δORs endogenously express on this type of cell. To verify this hypothesis, we separated and cultured the primary rat’s brain microvascular endothelial cells (BMECs), and then simultaneously immuno-stained the δOR and the vascular endothelial marker vWF. As Figure 3A shows, the immunofluorescent green signals representing δOR distributed evenly across all the cells, which co-localized with the red signals representing vWF. The opposite results were observed on the epithelial-derived cells HEK293, showing no expression of δOR (Appendix A). To further support this result, we also attempted to determine whether the protein of δOR could be detected by western blotting. As shown in Figure 3B, the primary cultured BMECs expressed endogenous δORs, while the HEK293 cells expressed few endogenous δORs. Taken together, we first showed the presence of endogenous δORs on BMECs.

### 3.4. DALDE Could Decrease OGD/R-Induced Apoptosis and Strengthen the Cell Viability of BMECs by Reducing the Cleaved Caspase-9 and the Ratio of Bax and Bcl-2

To verify the anti-apoptotic effect of DADLE on BMECs, we chose the oxygen–glucose-deprivation/reoxygenation (OGD/R) model to mimic the hypoxia-ischemic/reperfusion injury. According to Figure 4A,E, the apoptotic level of the BMECs significantly rose the cell viability that was largely weakened after 12 h of OGD, followed by 6 h of reoxygenation. The Bcl-2 and caspase families play an important part in the regulative process of apoptosis; Bcl-2 can inhibit apoptosis by a mitochondria-dependent caspase-9 pathway [34]. Bax protein translocates to the mitochondria and promotes endothelial cell apoptosis [35], exerting a pro-apoptotic effect against the action of Bcl-2 [36]. Therefore, the Bax/Bcl-2 protein ratio could be used to estimate the susceptibility status to apoptosis. As Figure 4C shows, an increased Bax/Bcl-2 protein ratio was tested in the BMECs exposed to OGD/R. Correspondingly, the protein level of cleaved caspase-9, which displayed the activation of caspase-9, ascended in the BMECs of the OGD/R group. These results showed the strengthened apoptosis of BMECs attributed to OGD/R. However, when we applied 5 nM DADLE during the process of hypoxia and reperfusion, the increased Bax/Bcl-2 protein ratio and level of cleaved caspase-9 were alleviated, and the administration of 10 nM DALDE markedly blocked these changes (Figure 4C,D). As a result, the degree of cellular apoptosis was abated (Figure 4B), and cell viability was recovered (Figure 4E) increasingly by DADLE in a dose-dependent manner. Additionally, 10 nM DADLE could evidently promote the proliferation of BMECs in normal conditions (Figure 4E). The above results show that DALDE could attenuate OGD/R-induced endothelial apoptosis and promote the survival of BMECs dependent on the Bax/Bcl-2/caspase-9 signaling pathway.

### 3.5. Both δOR Antagonist NTI and the Interference of δOR Could Block the Anti-Apoptosis and Pro-Survival Effects of DADLE on the OGD/R-Damaged BMECs

Next, to determine whether the participation of δOR is necessary for the peptide DADLE-induced anti-apoptosis and pro-survival effects on BMECs, we used δOR antagonist NTI to functionally block δOR and interfere with δOR’s expression by siRNA. As the TUNEL-staining results are shown in Figure 5A,B, the application of 10 nM DADLE, accompanied by 1 μM NTI, could completely reverse the anti-apoptosis action of DADLE. The DADLE-mediated relief of the damaged cell viability, caused by OGD/R, was also eliminated due to the existence of NTI (Figure 5C). On the other hand, if we downregulated the expression of δOR by 56.3% through siRNA interference (Figure 5D), the results of the TUNEL assay (Figure 5E,F) and CCK8 assay (Figure 5G) showed similar changes to the usage of NTI.

## 4. Discussion

In summary, this study discovers that delta opioid peptide DADLE targets BMECs, reducing apoptosis to relieve hypoxia-ischemic/reperfusion injury. In the animal model of MCAO/R, when pretreated with DADLE, cellular apoptotic specific diminishment focused on the cerebral vascular endothelial cells—but not the neurons, microglia, and astroglia—tissue injury, and how neurologic deficits can be relieved. In vitro, δORs’ expression was detected on the primarily cultured BMECs. Specifically, DADLE can reduce the cleaved caspase-9 and decrease the ratio of the protein levels of Bax and Bcl-2, resulting in decreasing apoptosis and the increasing viability of BMECs. These anti-apoptosis and pro-survival effects can be abolished by either δOR antagonist NTI or δOR siRNA transfection (Figure 6, made by BioRender).

The novelty of this study is that we first reported the presence of functionally expressed δOR on the primary BMECs (Figure 3). Compared to BMECs, the epithelium-derived cell line HEK293 showed no detection of δOR by immunofluorescence (Appendix A) or western blotting (Figure 3B), indicating that δOR exists expressed specifically in vascular endothelial cells. Whether δOR expresses merely in the cerebral vascular endothelial cells or differently in the vascular endothelial cells inside distinct organs could be ascertained in future work. This finding extends the range of δOR expression so that it is possible for δOR to show more physiological function participation. 

Second, we supplied new evidence that DADLE reduced the apoptosis of the cerebral vascular endothelial cells, possibly by activating their endogenous δORs in vivo rather than directly targeting neurons or glia. In the brain slices (Figure 2A) and in the primary BMECs (Figure 3A), DADLE treatment decreased the levels of apoptosis occurring in the cells being detected in the vascular endothelial cell’s marker vWF. DADLE also produced a pro-survival effect by downregulating the expression of Bax/Bcl-2, and cleaved caspase-9 to decrease cellular apoptosis and promote cell viability in vitro (Figure 4). On the other hand, if we blocked the binding of DADLE to δORs through giving NTI or downregulating the expression of δORs by transfecting targeted siRNA, the above effect was eliminated (Figure 5). Considering that DADLE can dramatically reduce the I/R injury of peripheral organs, such as the heart [37], liver [38], and kidney [39], it is necessary to further ascertain whether vascular endothelial cells are its targets. Additionally, some of our results confirmed the previous finding that DADLE could ease the I/R-mediated activation of astroglia [12]. According to the results shown in Appendix A, we inferred that the pretreatment of DADLE eased inflammation-related activation of microglia and astroglia, to some extent. 

Third, because of the special characteristics of the hippocampus tissue [40], its cells are more susceptible to I/R injury when compared to that of the cortex, which relates to I/R-induced spatial and long-term memory loss [41]. Correspondingly, in this study, it was easy to identify more apoptosis-positive objects in the dorsal hippocampus than in the cortex. However, in the areas of CA1, CA2, and DG, distributed apoptosis-positive signals, the signals that co-localized with vWF in the DG area, outnumbered those in the other subregions of the hippocampus or the cortex. Meanwhile, the co-localization of the neuron’s marker NeuN and apoptosis-positive signals appeared in the cortex the most. The apoptosis-positive microglia and astroglia also exhibited distributive differences (Figure 2F). These findings could support clues about brain area specificity for future studies concerning the pathophysiological process of cerebral I/R.

This study has a shortcomings and insufficiencies in the detailed signaling pathway exploration. Future work could focus on the potential key molecules in the DADLE-mediated anti-apoptosis effect. So far, the specific δOR signaling and mechanisms in preventing endothelial cell damage are less reported. However, it is indicated that the endogenous opioid peptides have stimulative effects of angiogenesis on the human umbilical vein endothelial cell (HUVEC), such as proliferation, migration, adhesion and tube formation [42]. One of the opioids, fentanyl, showed improved ischemic wounds healing in the rat [43]. Remifentanil enhanced the expression of PI3K/Akt/HIF-1α signaling in H/R-induced cardiac microvascular endothelial cell to rescue its dysfunction [44]. These findings all could support the idea that opioids have potential benefits for endothelial cells against ischemic injury. Thus, considering the maintenance of vascular endothelial cells’ survival is beneficial for the blood supply and integrity of the blood–brain barrier, whether DADLE affects the endothelium-dependent permeability of the capillary wall to improve the blood–brain barrier function can be a topic for future researchers to study. 

It is well known that the MCAO/R model and OGD/R could appropriately mimic the I/R process, respectively, in vivo and in vitro. We chose to pre-treat DADLE before the MCAO/R model or apply DADLE during the OGD/R situation, making it hard to apply this method for curing strokes after the occurrence; thus, tissue-protective agents during reperfusion should be further discovered. Several reports agree that the application of a delta opioid peptide during post-ischemia is beneficial [45,46], although opioids possess an addictive potency [47]. 

## Figures and Tables

**Figure 1 pharmaceutics-15-00046-f001:**
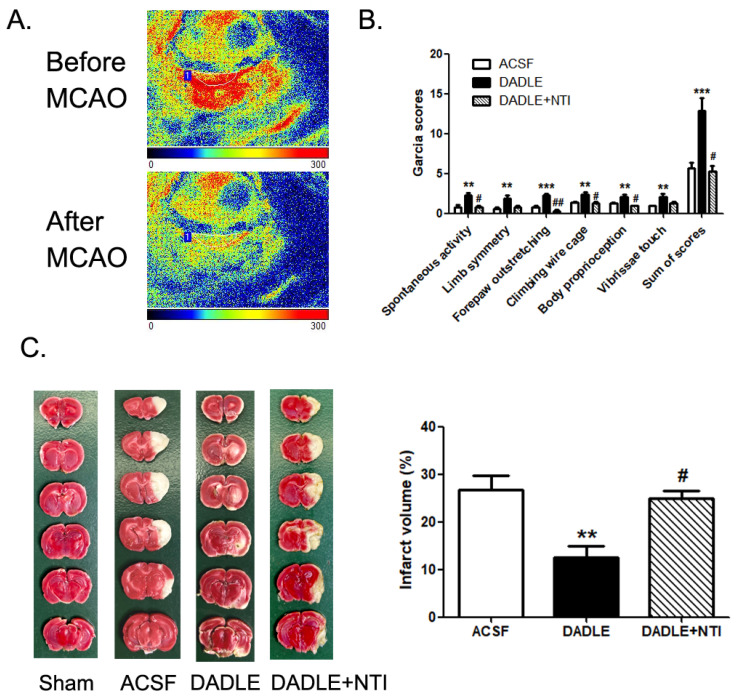
δOR agonist [D-ala^2^, D-leu^5^] enkephalin (DADLE) could reduce MCAO/reperfusion-induced cerebral ischemic infarct volume and relieve neurological deficit. (**A**) cerebral blood flow heatmap shows the hemisphere occlusive effect caused by MCAO surgery. (**B**) Garcia’s scores present the sensory and motor neurologic function of the MCAO-model-rats pre-treated with artificial cerebrospinal fluid (ACSF) or 5 nmol/10 μL DADLE or NTI (2.5 nmol/5 μL) following with DADLE at 72 h post-surgery. (**C**) 2,3,5-triphenyl tetrazolium chloride (TTC) staining shows the percentage of the cerebral infarct volume (showing as the white area) of the rats from the sham group and 72 h after MCAO/R with pretreatment of ACSF or 5 nmol/10 μL DADLE. The statistical column graph presents as mean ± S.E.M. ** *p* < 0.01, *** *p* < 0.001 versus the ACSF group. ^#^
*p* < 0.05, ^##^
*p* < 0.01 versus the DADLE group Both the ACSF and DADLE groups contained data from nine rats. The DADLE+NTI group contained data from four rats.

**Figure 2 pharmaceutics-15-00046-f002:**
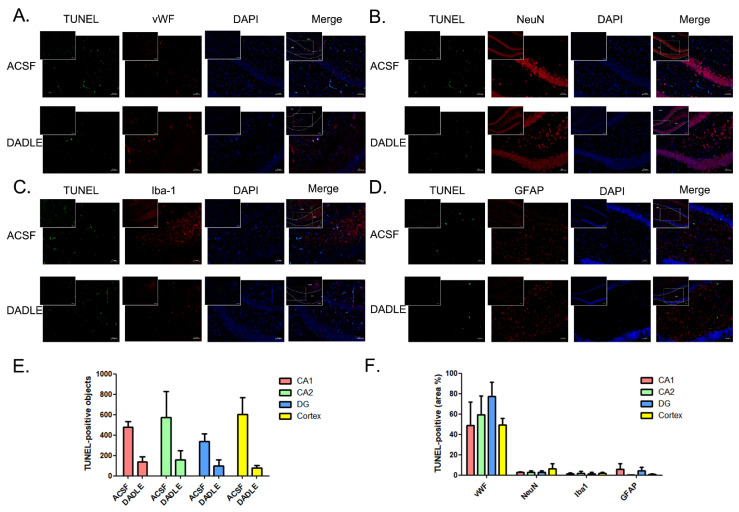
MCAO/R increased cellular apoptosis detected by TUNEL staining in the DG area of the hippocampus which mainly co-localized with vascular endothelial cells, but not neuron, microglia nor astroglia, and DADLE (5 nmol/10 μL) pretreatment decreased endothelial apoptosis at 72 h after surgery. (**A**–**D**) dUTP nick-end labelling (TUNEL)-positive cells were stained green. Nuclei were stained with DAPI and shown as blue. Red fluorescence signals represented different cellular markers. vWF was considered as the marker of vascular endothelial cells (**A**), NeuN as that of neuron (**B**), Iba-1 as that of microglia (**C**) and GFAP as that of astroglia (**D**). (**E**) Statistical analysis of the TUNEL-positive objects within the CA1, CA2 and DG of the hippocampus or cortex area of ACSF- and DADLE-pretreated group. (**F**) The percentage of TUNEL-positive signals within different brain areas. Bars represent the mean ± S.E.M. Every group contained data from three rats.

**Figure 3 pharmaceutics-15-00046-f003:**
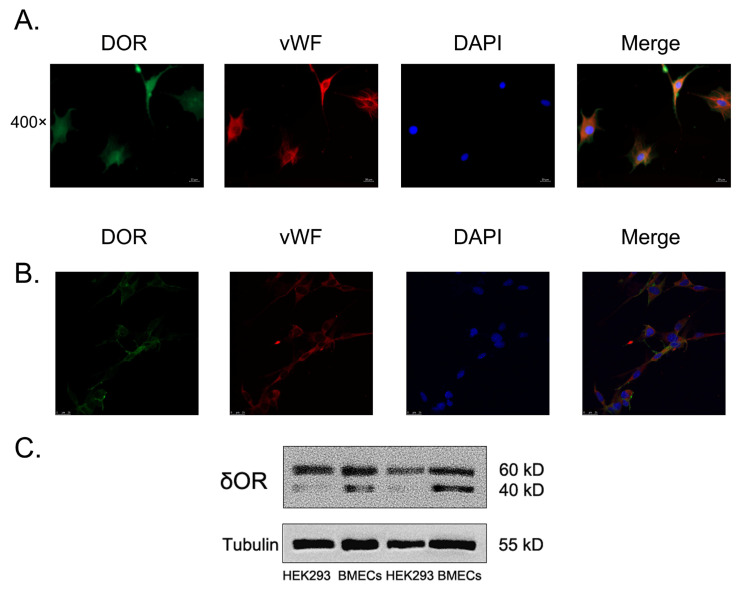
The primarily cultured BMECs express δORs. Typical immunofluorescence pictures (**A**) and confocal laser microscopy imaging (**B**) showed the expression of δORs (Green) and vWF (Red) on the BMECs. DAPI represented the nuclei and showed blue. (**C**) Western blotting detected the protein expression of δORs in the BMECs compared to that of HEK293 cells. Individual experiments were repeated three times.

**Figure 4 pharmaceutics-15-00046-f004:**
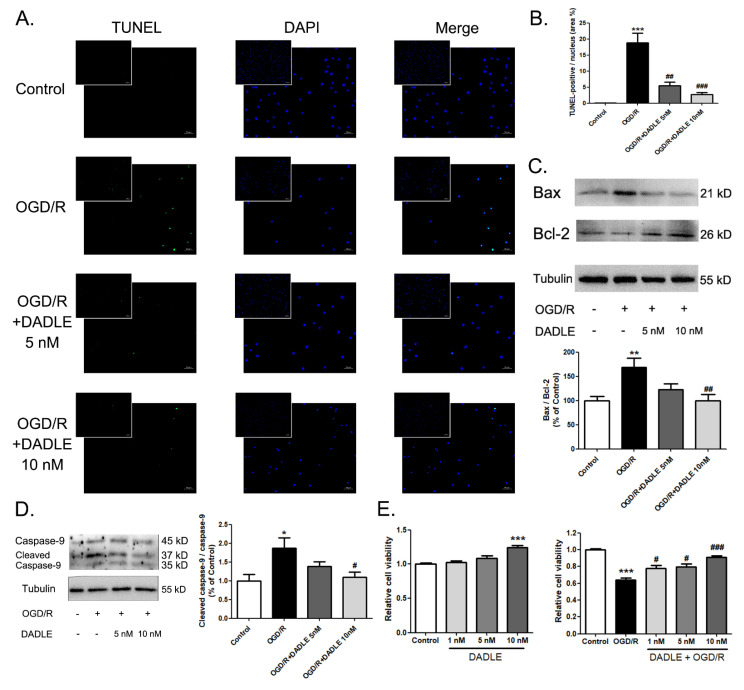
DADLE reduce apoptosis in the primarily cultured brain microvascular endothelial cells (BMECs) caused by the oxygen-glucose-deprivation/reoxygenation (OGD/R) via reducing the cleaved caspase-9 and the ratio of the protein level of Bax and Bcl-2 and then increase their viability in a dose-dependent manner. (**A**) TUNEL staining detection of OGD/R-treated BMECs with or without DADLE (5 nM or 10 nM). TUNEL-positive cells were stained green. Nuclei were stained with DAPI and showed as blue. (**B**) Statistical analysis of the TUNEL-positive nuclei (as the area percentage of total nuclei), n = 3. (**C**) Western blotting analysis of the expression of Bax and Bcl-2. Statistical analysis showed the ratio of Bax and Bcl-2, n = 8. (**D**) Western blotting analysis of the expression of caspase-9 and cleaved-caspase-9. Statistical analysis showed the ratio of cleaved-caspase-9 and caspase-9, n = 5. (**E**) CCK8 assay evaluated the relative cell viability after 12 h OGD following 6 h reoxygenation with or without DADLE, n = 4. Data were normalized compared to the control group. Bars represent the mean ± S.E.M.* *p* < 0.05, ** *p* < 0.01, *** *p* < 0.001 versus the control group. ^#^
*p* < 0.05, ^##^
*p* < 0.01, ^###^
*p* < 0.001 versus the OGD/R group.

**Figure 5 pharmaceutics-15-00046-f005:**
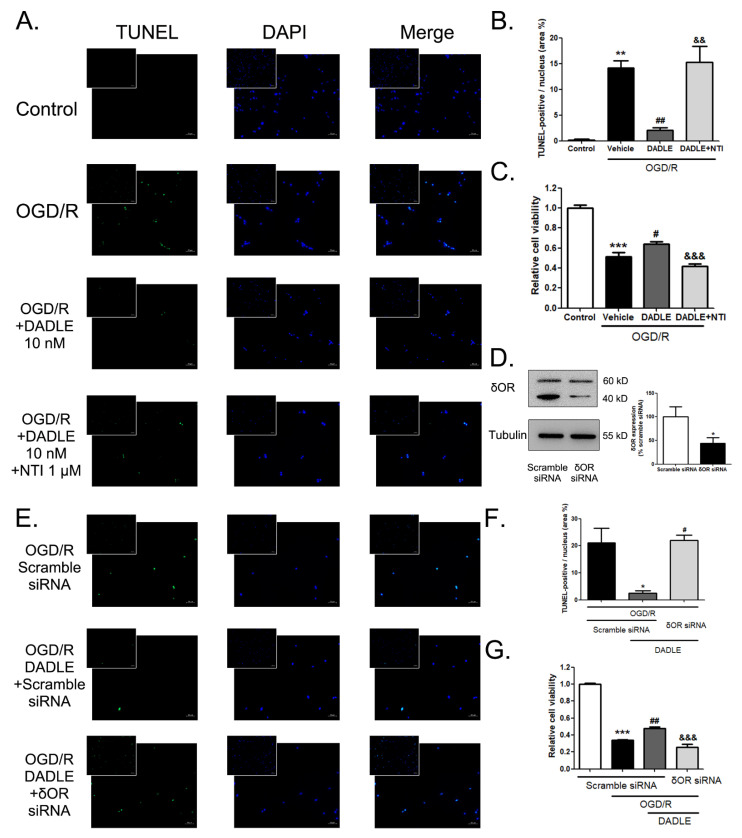
Both δOR antagonist NTI and interference of δOR can block DADLE-mediated decreased apoptosis in the OGD/R treated BMECs. (**A**) TUNEL staining detection of OGD/R-treated BMECs with 10 nM DADLE and an additional 1 μM NTI, and statistical analysis (**B**) showed the TUNEL-positive nuclei (as the area percentage of total nuclei), n = 4 or 5, ** *p* < 0.01 versus the control group; ^##^
*p* < 0.01 versus the OGD/R + vehicle group; ^&&^
*p* < 0.01 versus the OGD/R+DADLE group. TUNEL-positive cells were stained green. Nuclei were stained with DAPI and showed as blue. (**C**) CCK8 assay evaluated the relative cell viability of BMECs treated by 10 nM DADLE accompanied with 1 μM NTI after 12 h of OGD following 6 h reoxygenation. Data were normalized compared to the control group, n = 5, *** *p* < 0.001 versus the control group; ^#^
*p* < 0.01 versus the OGD/R+vehicle group; ^&&&^
*p* < 0.001 versus the OGD/R+DADLE group. (**D**) Western blotting detected the interfering effect of siRNA targeting δORs, n = 3, * *p* < 0.05 versus the scramble-siRNA-transfected group. (**E**) TUNEL staining detection of OGD/R-and-DADLE-treated BMECs accompanied with scramble siRNA or δOR siRNA transfection and statistical analysis (**F**) showed the TUNEL-positive nuclei (as the area percentage of total nuclei), n = 3 or 4, * *p* < 0.05 versus the scramble-siRNA-transfected group; ^#^
*p* < 0.01 versus the DADLE-treated scramble-siRNA-transfected OGD/R group. (**G**) CCK8 assay evaluated the relative cell viability of BMECs treated by 10 nM DADLE accompanied with scramble siRNA or δOR siRNA transfection after 12 h OGD following 6 h reoxygenation. Data were normalized compared to the scramble siRNA group, n = 4, *** *p* < 0.001 versus the scramble-siRNA-transfected group; ^##^
*p* < 0.01 versus the DADLE-treated scramble-siRNA-transfected OGD/R group; ^&&&^
*p* < 0.001 versus the DADLE-treated δOR-siRNA-transfected OGD/R group. Bars represent the mean ± S.E.M.

**Figure 6 pharmaceutics-15-00046-f006:**
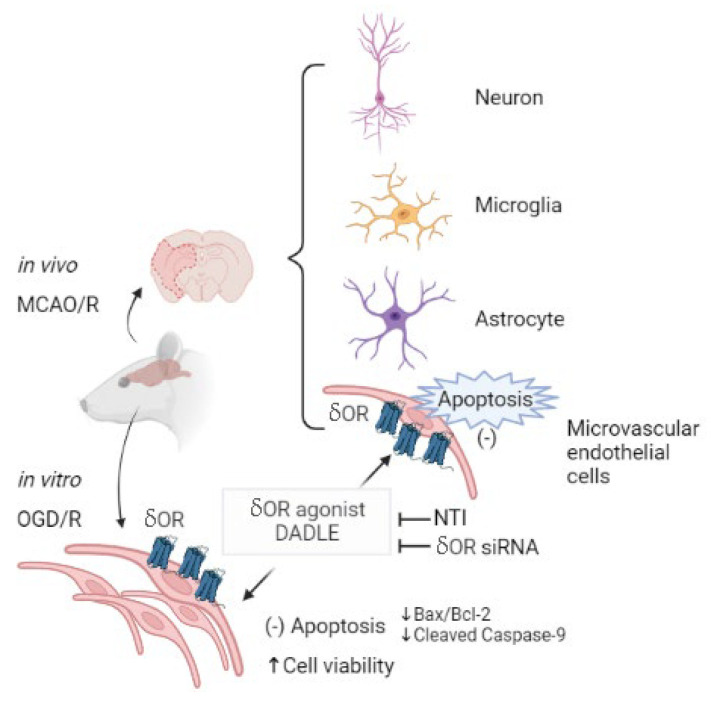
Schematic representation shows that delta opioid peptide DADLE targets BMECs reducing apoptosis to relieve hypoxia-ischemic/reperfusion injury. Pre-treatment with the δOR agonist DADLE can reduce cerebral vascular endothelial apoptosis and then relieve tissue injury induced by MCAO/R. In vitro, δORs’ expression is detected on the primarily cultured BMECs. DADLE can reduce the apoptosis of BMECs, and then increase their viability. Both δOR antagonist NTI and interference of δOR can abolish this effect. Specifically, DADLE can reduce the cleaved caspase-9 and decrease the ratio of the protein level of Bax and Bcl-2.

## Data Availability

The data generated and analyzed in this study are available from the corresponding author on request.

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
