# Peer review of "Delta Opioid Peptide Targets Brain Microvascular Endothelial Cells Reducing Apoptosis to Relieve Hypoxia-Ischemic/Reperfusion Injury"

_pharmaceutics, 2022, doi:10.3390/pharmaceutics15010046_

Round 1
Reviewer 1 Report
I found this manuscript real interesting, clear and well written. The text is well formulated – I’m far not native speaker, and therefore maybe somebody would find mistakes, but I haven’t got problems with the text. And the most important I found all the date valid without any tricks. I would have only some minor comments and questions:
1. In chapter 2.3 it was mentioned that 21% of the animals were lost during the MCAO/R model. Were they ca. equally died in both control and DADLE group?
2. In chapter 2.4 first the whole in vivo evaluation process was little bit superficial, but in the original paper of Garcia et al. I found the whole method, so it is acceptable in this form.
3. In chapter 2.5 please clear what kind of anesthetic was used! Also here: how many slices were made and evaluated / rat brain, and how many sham animals were used?
4. In chapter 3.2 it was shown in control rats the co-localization of TUNEL positive objects with different cellular markers in different regions (Fig2E), the effect of DADLE was only show generally without region and cellular marker details (Fig2F). Please show the effect of DADLE detailed as it was made in controls! That should be included in the final version.
5. Also in chapter 3.2 was mentioned that some morphological changes on microglia and astroglia was observed. Please quantify this observation!
6. In Fig4 it is shown that 10nM DADLE practically normalized cell viability and almost totally reduced apoptotic markers. In contrast in Fig5 the same concentration of DADLE without NTI or DOR siRNA was much less effective. Please clear this discrepancy!
So altogether this manuscript needs only minimal improvement and it can be accepted thereafter.
Author Response
We appreciate the editor’s and reviewers’ efforts in evaluating our manuscript. Responses to each reviewer are as followed. Here we upload the point-to-point response to reviewer 1, the revised manuscript and figures. Please see the attachment file.

Reviewer 2 Report
In this manuscript by Zhang and colleagues, provide data about DOR agonist DADLE mediated preventive effect of endothelial cell damage in a brain I/R injury model. They also tested the DOR agonist DADLE in in-vitro hypoxia-ischemia/reperfusion model in BMECs. They showed that DADLE can reduce I/R injury mediated endothelial cell apoptosis. There are several comments that need to be addressed along with minor edits that would clarify and strengthen the manuscript and its conclusions.
1. Authors tested neuroprotective effects of DADLE against brain I/R injury in in-vivo and in-vitro I/R injury model in isolated BMECs. Subject and results are interesting. However, they did not test the specificity of DOR agonist DADEL by using DOR antagonist NTI as in in-vitro studies on BMECs. Without these experiments the effects observed with DADLE is not conclusive about DOR-activity mediated event in-vivo.
2. In Fig5A DOR activation prevented TUNEL + cell number in I/R injury model. In Fig5E DOR expression is ~50% downregulated with siRNA approach but it is not clear which of the two bands of DOR they calculated (or both). In Fig 5E, did they perform I/R injury model as the results of TUNEL assay and cell viability measurements given in Fig5F, G? Comparing the data in Fig 5C and G, cell viabilities in vehicle and scramble siRNA from I/R injury model are different. Also, the efficiency of DADLE reversing the decreased cell viability is different. DADLE activation in DOR silenced cells does not have dramatic effect in cell viability though statistically significant. If DOR activation would be required for preventing I/R injury in endothelial cell survival, then activation of DOR in silenced cells would have more detrimental effects on cell viability. However, it has minor effect. Authors do not argue about this finding. Can this finding be due to any change in the cell surface expression of DOR during I/R injury in BMECs because 12h is long enough to cause desensitization along with long ischemia? Similarly, how I/R injury in-vivo would change the expression of DOR? This is important for the effectivity of DADLE treatment in a potential clinical application.
3. Discussion should give more information about DOR action/signaling and mechanisms in preventing endothelial cell damage. Also, last part of the discussion has overstatements about DADLE use and clinical applications in preventing stroke.
4. In Middle Cerebral Artery Occlusion and Reperfusion experiments ischemia was done for 45 min and followed by 1h of reperfusion. In Oxygen Glucose Deprivation/Reperfusion experiments cells were kept in anoxia for 12h + glucose free media and then reperfused for 6h. In latter there are many changes occurring even at the transcriptional level besides other metabolic adaptive changes. The major concern is does in-vitro experiments really reflect the in-vivo data.
5. Which cells have been used for viability assay? Cells obtained from MCAO/R rats or healthy rats?
6. In Fig4C Bax-Bcl2 WB bands are distorted. Original images don't show excellent quality to make analysis. In Fig1B statistical significances has been only shown for total scores, but not for each neurological parameter.
7. Abbreviations (e.g. line 69), percentiles (e.g. line 44), numbers (e.g. line 83) in sentences should be used properly.
Minor Comment:
Major findings of the study given at the end of introduction section should be shortened. Similarly, methods section should be shortened.
In Fig.5 legend stat analysis results should be given for all data.
Author Response
We appreciate the editor’s and reviewers’ efforts in evaluating our manuscript. Responses to each reviewer are as followed. Here we upload the point-to-point response to reviewer 2, the revised manuscript and figures. Please see the attachment file.

Round 2
Reviewer 2 Report
Authors have addressed all my questions and included in the text. I do not have further comments. A minor point is e.g. line 41: sentence should not start with number.